METHODS

# teemi: An open-source literate programming approach for iterative design-build-test-learn cycles in bioengineering

Søren D. Petersen[1‡], Lucas Levassor[1,2‡], Christine M. Pedersen[1], Jan Madsen[3], Lea G. Hansen[1], Jie Zhang[1], Ahmad K. Haidar[1], Rasmus J. N. Frandsen[2], Jay D. Keasling[1,4,5,6,7], Tilmann Weber[1], Nikolaus Sonnenschein[2¤], Michael K. Jensen[1] *

**1** Novo Nordisk Foundation Center for Biosustainability, Technical University of Denmark, Kgs. Lyngby, Denmark, **2** Department of Biotechnology and Biomedicine, Technical University of Denmark, Kgs. Lyngby, Denmark, **3** Department of Applied Mathematics and Computer Science, Technical University of Denmark, Kgs. Lyngby, Denmark, **4** Joint BioEnergy Institute, Emeryville, California, United States of America, **5** Biological Systems and Engineering Division, Lawrence Berkeley National Laboratory, Berkeley, California, United States of America, **6** Department of Chemical and Biomolecular Engineering, Department of Bioengineering, University of California, Berkeley, California, United States of America, **7** Center for Synthetic Biochemistry, Institute for Synthetic Biology, Shenzhen Institutes of Advanced Technologies, Shenzhen, China

¤ Current address: Ginkgo Bioworks, Boston, Massachusetts, United States of America
‡ These authors share first authorship on this work.
* mije@biosustain.dtu.dk

**Data Availability Statement:** Data availability statement The paper and its supplementary information files provide data that support the

## Abstract

Synthetic biology dictates the data-driven engineering of biocatalysis, cellular functions, and organism behavior. Integral to synthetic biology is the aspiration to efficiently find, access, interoperate, and reuse high-quality data on genotype-phenotype relationships of native and engineered biosystems under FAIR principles, and from this facilitate forward-engineering strategies. However, biology is complex at the regulatory level, and noisy at the operational level, thus necessitating systematic and diligent data handling at all levels of the design, build, and test phases in order to maximize learning in the iterative design-build-test-learn engineering cycle. To enable user-friendly simulation, organization, and guidance for the engineering of biosystems, we have developed an open-source python-based computer-aided design and analysis platform operating under a literate programming user-interface hosted on Github. The platform is called teemi and is fully compliant with FAIR principles. In this study we apply teemi for i) designing and simulating bioengineering, ii) integrating and analyzing multivariate datasets, and iii) machine-learning for predictive engineering of metabolic pathway designs for production of a key precursor to medicinal alkaloids in yeast. The teemi platform is publicly available at PyPi and GitHub.

## Author summary

Bioengineering holds fantastic perspectives and is poised to change how we produce foods, materials, and medicines. However, rapid progress is limited by a lack of

findings of this study. All data related to this study can be accessed and downloaded from GitHub, the designated data repository at https://github.com/hiyama341/G8H_CPR_library. The data include all source files and datasets analyzed throughout the study as well as training sets for the machine-learning models. Code availability The code utilized for data extraction, organization, filtering, and simulation, as well as the code utilized for algorithm training, can be found on GitHub https://github.com/hiyama341/teemi. The teemi platform was implemented through PyPi and is available at https://pypi.org/project/teemi/.

**Funding:** This work was supported by Novo Nordisk Foundation Center for Biosustainability grant number NNF20CC0035580 and by the European Union Horizon 2020 research and innovation program grant agreement number 814645 (MIAMi) to M.K.J. N.S. acknowledges funding from the Novo Nordisk Foundation under the Fermentation Based Biomanufacturing program (grant no. NNF17SA0031362). URLs: https://novonordiskfonden.dk/grant/ and https://ec.europa.eu/info/funding-tenders/opportunities/portal/screen/programmes/horizon The funders had no role in study design, data collection and analysis, decision to publish, or preparation of the manuscript. Salary of S.P. was funded by European Union Horizon 2020 research and innovation program grant agreement number 814645 (MIAMi).

**Competing interests:** The authors have read the journal's policy and the authors of this manuscript have the following competing interests: M.K.J., L.G.H. J.D.K. and J.Z. are inventors on pending patent applications. M.K.J., L.G.H., J.D.K., and J.Z. have financial interest in Biomia Aps. J.D.K. also has a financial interest in Amyris, Lygos, Demetrix, Napigen, Apertor Pharmaceuticals, Maple Bio, Ansa Biotechnologies, Berkeley Yeast, and Zero Acre Farms. The remaining authors declare no competing interests.

mechanistic knowledge in even the simplest model organisms, such as bacteria and yeast. Thus, to compensate, we often have to construct and study a large number of engineered cells and select the cells with the greatest potential for the given objective function. The targeted construction of engineered cells is often described as an iterative process of design, build, test, and learn (the DBTL cycle). Literate programming is a paradigm that encourages the combination of text and computer code with the potential to describe all workflows covered by the DBTL cycle. The purpose of the present work is to give a first estimate of the extent to which we can accelerate the individual steps in the DBTL cycle by using end-to-end literate programming workflows. To achieve this, we established an open-source platform called teemi, and used it to optimize production of a key precursor to medicinal alkaloids in yeast. We expect that teemi will enable higher DBTL throughput with fewer errors, better integration of IT tools with laboratory resources, and more effective knowledge capture.

This is a *PLOS Computational Biology* Methods paper.

## Introduction

The rational engineering of biology for user-defined purposes, also known as synthetic biology, has fostered a shift in the way we imagine, design and produce foods, materials, and medicines [1]. Seminal examples of synthetic biology success stories adopted by society during the last decade includes plant-based burgers with meat flavor derived from soy leghemoglobin produced in engineered yeast [2], the hyaline family of clear, flexible and robust polyimide films for flexible electronics made from bio-sourced monomers [3], and chimeric antigen receptors (CARs) fused to antibodies that when inserted in patients' T cells and introduced into the patient enable efficient killing of cancer cells [4]. Common to these examples is the bioengineering of living cells to encapsulate and arm them with novel functions to meet societal needs in agriculture, manufacturing industry, and health. Even more so, many more solutions to mitigate climate changes, increase food supplies, and treat patients with unmet needs are set to depart from engineered cells and synthetic biology in the near future [5].

However, rapid progress in bioengineering is limited by the long, costly, and non-standardised approaches used to engineer even the simplest model cells, such as *Escherichia coli* and *Saccharomyces cerevisiae* [6]. Taken together with the molecular and metabolic complexity of biological systems, and limited scalable design principles, bioengineers often have to construct and study large libraries of variant cell designs to identify genotypes with sought-for properties [7]. The targeted construction of strains is often described as an iterative process of design, build, test, and learn (the DBTL cycle) [7]. To support the various steps of the DBTL cycle a multitude of commercial software and cloud-lab platforms are available, including Benchling, Riffyn, Inscripta, Teselagen and Emerald Cloud Lab, with advanced laboratory information management system (LIMS), data analysis capabilities, and integration of laboratory workflow execution via robotics [8,9]. In addition to commercial platforms, open-source Python APIs for flexible workflow planning, execution and data management, central to the working practices of researchers, are gaining momentum, especially covering the design and learn steps of the DBTL cycle [10–12] Similarly, the collection of FAIR (Findable, Accessible, Interoperable,

Reusable) training materials made available via community efforts, such as the Galaxy Training Network [13], seek to empower researchers with data analysis literacy and bridge the skills gap between design-build-test and learn [14]. Still there is a common challenge to further support researchers in using natural language laboratory protocols and integrating such tools and services into their daily workflows [15]. Solving this challenge should enable i) that more tasks can be performed in a shorter amount of time and with less errors, ii) better integration of IT tools with other laboratory resources, such as robotics, and iii) better documentation, and thus more effective knowledge transfer among research communities [15–17].

Literate programming is a paradigm that encourages the combination of text and computer code in a systematic and coherent way [18]. Computer code is formal language for describing how to do things [19]. The code can be understood by both humans and computers if it is written sufficiently abstract. Literate programming protocols are thus written for humans, but computer code is used whenever the tasks can be performed by a computer. With literate programming, workflows and data can be described precisely meeting the FAIR principles [20].

In particular within bioengineering, a literate programming approach has the potential to describe all elements of the DBTL cycle, thus supporting cost-effective laboratory and data analytical workflows. Examples of existing open-source tools that excel in parts of the DBTL cycle include Pydna [21], Aquarium [11], GalaxySynBioCAD [22]. Briefly, Pydna provides descriptions of DNA assembly and cloning strategies in Python with a high degree of flexibility. Complementary to this, Aquarium generates protocols represented as executable code with an integrated LIMS system, while the Galaxy platform [23] offers extensive bioinformatic workflows on genomic analysis through its web-based platform. Extending from this, Galaxy-SynBioCAD builds on the Galaxy platform with a focus on tools for synthetic biology, such as retrosynthesis and metabolic pathway analyses, through the DBTL cycle. Based on this, there is still a need to provide examples of how tools like these can be assembled into flexible end-to-end DBTL workflows leveraging the best of an ever-increasing palette of tools through literate programming.

The purpose of the present work is to give a first estimate of the extent to which bioengineers can accelerate the speed, efficiency, and fidelity of the individual steps in the DBTL cycle by using literate programming. To do so, we have established an open-source platform including all elements of the iterative DBTL cycle bioengineers are confronted with. The platform is called teemi. To showcase teemi in its entirety and facility efficient adoption, we present an experimental example using literate programming in teemi for all DBTL stages of an iterative learning task targeting the optimization of a metabolic pathway for production of a key precursor to medicinal alkaloids in yeast.

## Results

### Background and motivation for teemi

At the onset of this project, we first assessed the availability of web tools and scripts available for bioengineers to streamline DBTL workflows (Table 1). While tools like GalaxySynBioCad and Aquarium are widely adopted [9,11], we could not identify open-source tools that can integrate all steps of the DBTL cycle in a single workflow, without the need to acquire programming skills, and/or shifting between platforms and programming languages.

In literate programming, besides the textual documentation, embedded code allows abstracting away all computations in a reusable way. Lab notebook-style chronological documents will contain information on when, how, and for what purpose, data was acquired and used. Moreover, with literate programming, data is compliant with FAIR principles being findable and accessible from a single context via links to digital repositories, interoperable via a

**Table 1. Comparison of maintained\* open-source IT tools and their functionalities for full-stack DBTL cycle.**

| | Pydna | Aquarium | Galaxy platform | Galaxy SynBioCAD | Poly | Edinburgh Genome Foundry tools | Lattice-Automation/synbio | autoprotocol-python |
|---|---|---|---|---|---|---|---|---|
| **DESIGN:** Parts selection | - | - | - | + | - | + | - | - |
| **DESIGN:** Combinatorial library generation | - | - | - | + | - | - | - | - |
| **DESIGN:** Cloning workflows | + | + | - | + | + | - | + | - |
| **BUILD:** Laboratory protocols | - | + | - | - | + | - | - | + |
| **BUILD:** Automation with robotics | + | - | - | + | - | + | + | - |
| **TEST:** Data processing of analytics | - | + | + | + | - | - | - | - |
| **LEARN:** Machine-Learning | - | - | + | - | - | - | - | - |
| **LIMS system** | - | | - | - | - | - | - | - |
| **Python level** | Medium | None | None | None | Go-package | Medium | Medium | Medium |

\* Minimum one commit on GitHub within the last year.

free to use, open source, and user-friendly workflow document, while both data acquisition and processing are reproducible via text documentation and embedded functions.

For this study, teemi is used in Jupyter Notebooks and consists of a set of Python functions and classes facilitating simulation of experimental flow for *in vivo* design and assembly of diverse genetic libraries, pooled library constructions, organization and modeling of genotype and phenotype data, as well as implementing machine learning to model the data and recommend new designs (Figs 1 and S1). Through teemi simulations, the preparation of laboratory work is standardized and thoroughly executed, aimed at reducing time consumption, decreasing human error rates, and improving the reproducibility of experimental results.

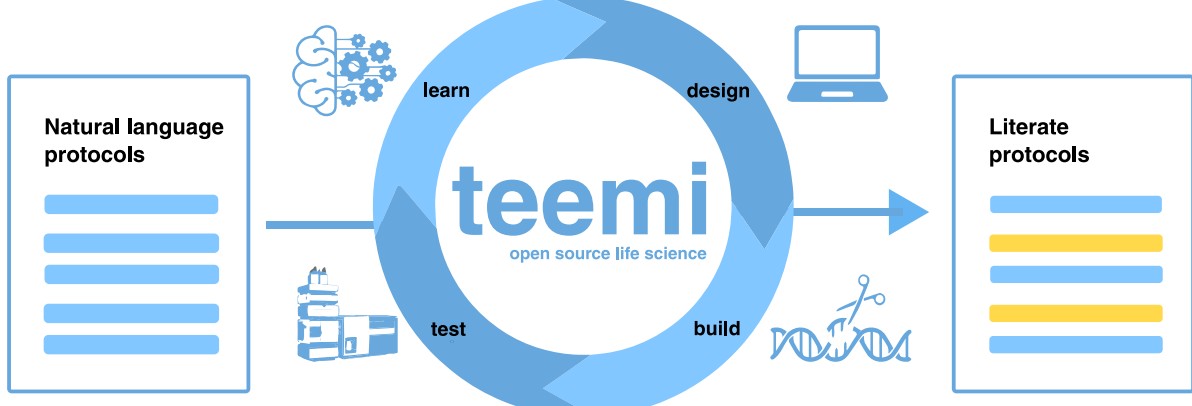

**Fig 1. Conversion of natural language lab protocols for iterative design-build-test-learn cycles to literate protocols using teemi.** Natural language protocols (left—blue) comprehensible to humans are converted into computer code (right—yellow) that can be understood by both computers and humans. In teemi, each procedure in natural language protocols is connected with names of python modules in literate protocols, thus lowering the programming entry level needed for adopting teemi. See also S1 Fig for more details. Created with Biorender.com.

**Table 2. Overview of the notebooks created for this work.**

| DBTL Round | | Name and link | Description |
|---|---|---|---|
| **1** | **DESIGN** | 00_1_DESIGN_Homologs | Describes how we automatically can select homologs from NCBI from a query in a standardizable and repeatable way. |
| | | 01_1_DESIGN_Promoters | Describes how promoters can be selected from RNAseq data and fetched from an online database with various quality measurements implemented. |
| | | 02_1_DESIGN_Combinatorial_library | Describes how a combinatorial library can be generated with the DesignAssembly class along with robot executable instructions. |
| | **BUILD** | 03_1_BUILD_gRNA_plasmid | Describes the assembly of a CRISPR plasmid with USER cloning. |
| | | 04_1_BUILD_Background_strain | Describes the construction of the background strain by K/O of G8H and CPR in the X-3 and XI-3 sites respectively. |
| | | 05_1_BUILD_Combinatorial_library | Building a combinatorial library of 1280 combinations with designs generated by Tesselagen software. |
| | **TEST** | 06_1_TEST_Library_characterisation | Describes data processing of LC-MS data and genotyping of the generated strains. |
| | **LEARN** | 07_1_LEARN_Modelling_and_predictions | Describes the use AutoML to predict the best combinations for a targeted second round of library construction. |
| **2** | **DESIGN** | 08_2_DESIGN_Model_recommended_combinatiorial_library | This notebook utilizes the machine learning predictions made in the previous notebook to create a targeted combinatorial library with best predicted genetic parts. |
| | **BUILD** | 09_2_BUILD_Combinatorial_library | Shows how results from the ML can be translated into making a second focused library of strains. |
| | **TEST** | 10_2_TEST_Library_characterization | Describes the data processing of LC-MS data like in notebook 8 but with the second focused library. |
| | **LEARN** | 11_2_LEARN_Modelling_and_predictions | Second cycle of ML showing how the model increased performance and saturation of best-performing strains. |

The literate programming notebooks used for the experimental testbed presented in this study are hosted by Google Colab. All notebooks are extensively referenced upon implementation throughout this study as well as summarized in a comprehensive list (Table 2), allowing the reader to easily connect literate programming for iterative DBTL cycles with the results presented.

## The experimental bioengineering testbed

An often-encountered bottleneck in modern biotechnology is the bottleneck of oxidation reactions catalyzed by cytochrome P450 enzymes [24,25]. These oxidation reactions are catalyzed by cytochrome P450 (CYP) superfamily of hemoproteins, and cytochrome P450 reductases (CPR) [26–28]. CYPs are generally often speaking cytosol-facing, N-terminally bound enzymes bound to the endoplasmic reticulum (ER) [24]. They catalyze hydroxylations of small molecule substrates facilitated by the transfer of two electrons from NADPH to NADP+ catalyzed by the ER-bound CPRs [27]. Plant-derived CYP/CPR reactions are widespread in modern biotechnology for fermentation-based manufacturing of fine chemistries, such as alkaloids and terpenes [29,30]. When heterologously expressed in microbes, such as the biotechnology workhorse baker's yeast *Saccharomyces cerevisiae*, poor CYP activity and shunt product formation limits efficient bioconversion of cheap feedstocks to value-added advanced pharmaceutical ingredients sourced by fermentation [25,29]. To mitigate this, CYP/CPR reactions often need extensive trial-error engineering to optimize substrate conversion and balance co-factor availability in cell factories. This has included i) regulating the expression of genes encoding both CPR and CYPs, ii) searching for optimal CYP:CPR pairs, iii) bioprospecting for enzyme homologs, iv) perturbing gene copy numbers, or v) rational engineering of signal peptides to target membrane-anchoring of enzymes to dedicated subcellular compartments [25,27]. While independently all of these approaches have positively impacted oxidation reactions catalyzed

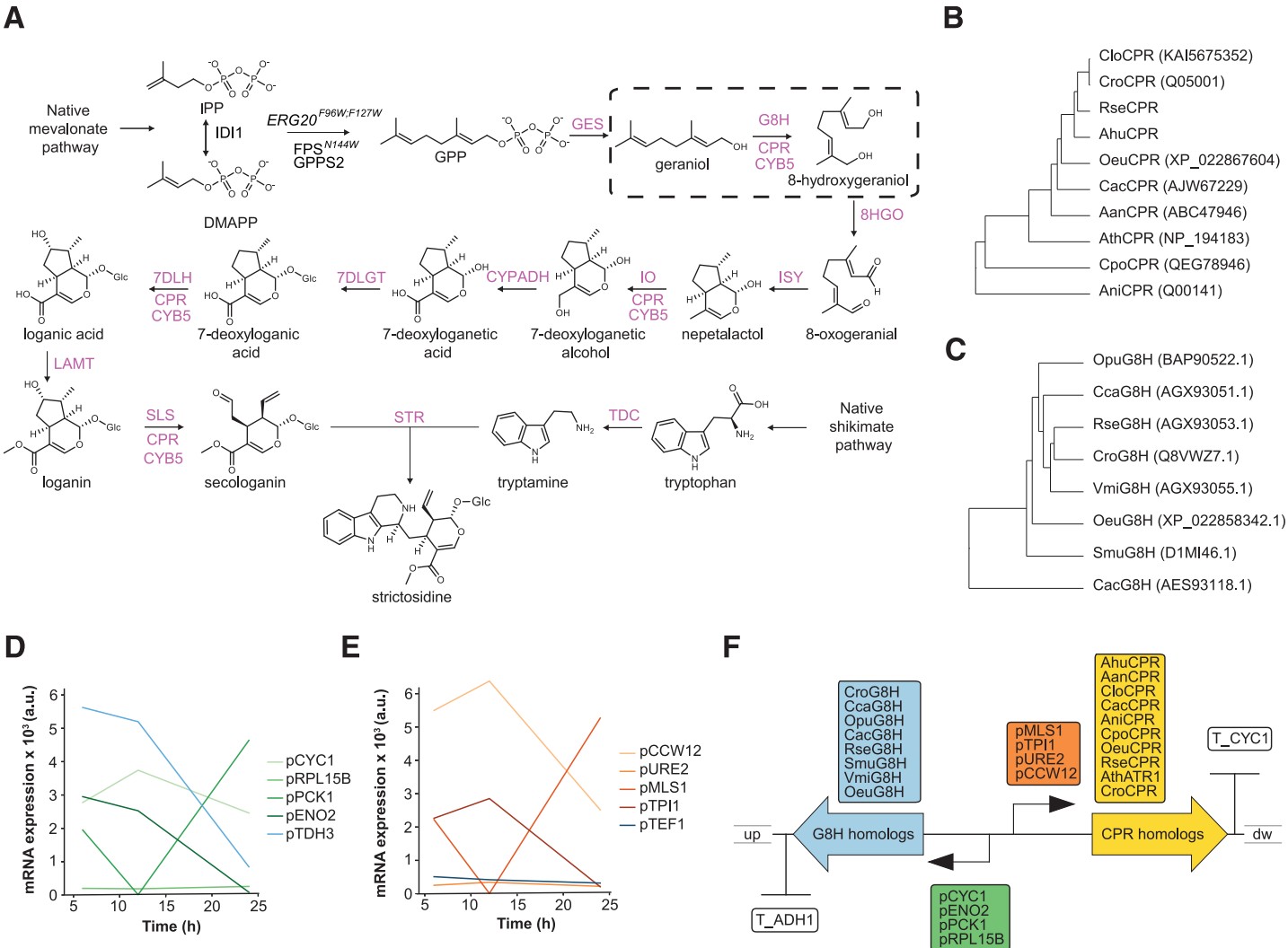

**Fig 2. Design and characteristics of the constituent DNA parts used as experimental testbed for teemi.** (**A**) The ten-step biosynthetic pathway converting geraniol to strictosidine. The G8H step is highlighted in a dashed box [26]. (**B-C**) Rooted phylogenetic trees of G8H (D) and CPR (E) protein representatives. Uniprot identifiers are shown in parentheses. Catharanthus roseus (Cro), Rauvolfia serpentina (Rse), Olea europaea (Oeu), Camptotheca acuminata (Cac), Vinca minor (Vmi), Cinchona calisaya (Cca), Ophiarrhiza pumila (Opu), and Swertia mussatii (Smu), Artemisia annua (Aan), Arabidopsis thaliana (Ath), Catharanthus longifolius (Clo), Amsania hubrichtii (Ahu), and Aspergillus niger (Ani). (**D-E**) Temporal resolution of transcript abundances for candidate genes [34], for which promoters were chosen to control the expression of genes encoding G8H (D) and CPR (D) homologous. (**F**) Combinatorial assembly and genome integration strategy.

by heterologous expression of plant-derived CYPs and CPRs in yeast [25,27,29], multivariate exploration of these reactions are needed. One recent study documenting the power of combinatorial search strategies was performed by Davies *et al.*, searching >100 CYP/CPR co-expression designs, which when combined with best-performing promoter designs show-cased improved C8-hydroxylation of geraniol to 8-hydroxy-geraniol catalyzed by the geraniol hydroxylase G8H and its CPR partner [27].

In this study we present the power of teemi and literate programming to build simulation-guided and iterative laboratory workflows for optimizing strictosidine production in yeast (Fig 2A). Motivated by the complexity of the oxidation reactions and documented importance of exploring combinatorial design spaces [27], and the observation that feeding 8-hydroxy-geraniol improves strictosidine production compared to feeding geraniol [29], we considered the

C8-hydroxylation of geraniol to 8-hydroxy-geraniol as a valid testbed to showcase the bandwidth and throughput enabled by literate programming using teemi.

## teemi for design-build-test-learn cycle I

Using teemi we initially constructed a parental strain (MIA-CH-A2) harboring *Cro*G8H and *Cro*CPR under the control of promoters pTDH3 and pTEF1, together with the other 11 genes controlling the expression of genes encoding enzymes of the biosynthesis pathway converting geraniol to strictosidine [26,29](Fig 2A).

The first iteration of the teemi-based DESIGN module, under the DBTL framework, focused on enzyme homology searches, promoter choices, and primer designs. This approach aimed to leverage the extensive and increasing genomic resources available in databases like NCBI [31]. Using a top-down approach is particularly useful when the enzymatic pathway is known. Building on this, we developed an algorithm to standardize screening and selection of homologs (00_1_DESIGN_Homologs, Paragraph: 1) using Catharanthus roseus G8H and CPR sequences as queries [23,25].

In addition to the NCBI database search, CPR candidates documented from literature [29,32], search results from the PhytoMetasyn database [33], and a beetle G8H from *Chrysamela populi* (*Cpo*) were included to generate diversity (Fig 2B and 2C).

Each gene was expressed under the control of four unique native promoters, yielding a total library size of 1,280 (8x10x4x4). For the choice of promoters a second algorithm was developed aimed at selecting relevant promoters from expression data generated during the lag (10% glucose consumption, low ethanol production), mid-exponential (75% glucose consumed, increasing ethanol production), and post-exponential phases (>99% glucose consumed, start of ethanol consumption) [34](01_1_DESIGN_Promoters, **Paragraph: 1**)). All promoter sequences were aligned to ensure that there were no homologous sequences in order to minimize recombineering during transformation and library propagation (01_1_DESIGN_Promoters, **Paragraph: 4**). Lastly, primers for amplification of each of the chosen library parts were designed (02_1_DESIGN_Combinatorial_library, **Paragraph: 3**). To facilitate homologous recombinations by design, the parts used as flanking regions for repair assembly into a pre-defined genomic landing pad were designed to be 0.5 kb and the homology regions between library parts were 30 bp by default. For all design steps, the notebooks along with teemi were used to simulate all relevant designs in a combinatorial library, check primer matches with templates, calculate lengths of PCR products, and print tables of PCR mixes in order to provide an overview of reagents and their location, calculate melting temperatures for PCR programs and expected gel electrophoresis outputs, and create expected sequences from an alignment of parts integrated (03_1_BUILD_gRNA_plasmid, 04_1_BUILD_Background_strain, 05_1_BUILD_Combinatorial_library, 09_2_BUILD_Combinatorial_library). As such this simulation also mimics an electronic laboratory notebook (ELN), thus facilitating documentation of the experiments and allowing for easy sharing in order to prevent knowledge loss. Most importantly, the 100% sequence verification of amplicons (06_1_TEST_Library_characterisation, **Paragraph: 2.2**; 10_2_TEST_Library_characterization, **Paragraph: 2**) based on teemi simulations of expected gel electrophoresis outputs (S2 Fig) is a validation of the simulation workflow, and is expected to improve interoperability and reproducibility of laboratory workflows, and help reduce human errors.

Next, for the BUILD module, we adopted CasEMBLR for CRISPR/Cas9-mediated assembly harnessing seamless homologous recombination between seven parts in each cluster [35], and into a stable genomic integration site [36] (Fig 3A). The choice of method used for the BUILD step, focused on reducing potential expression heterogeneity of the designed expression units

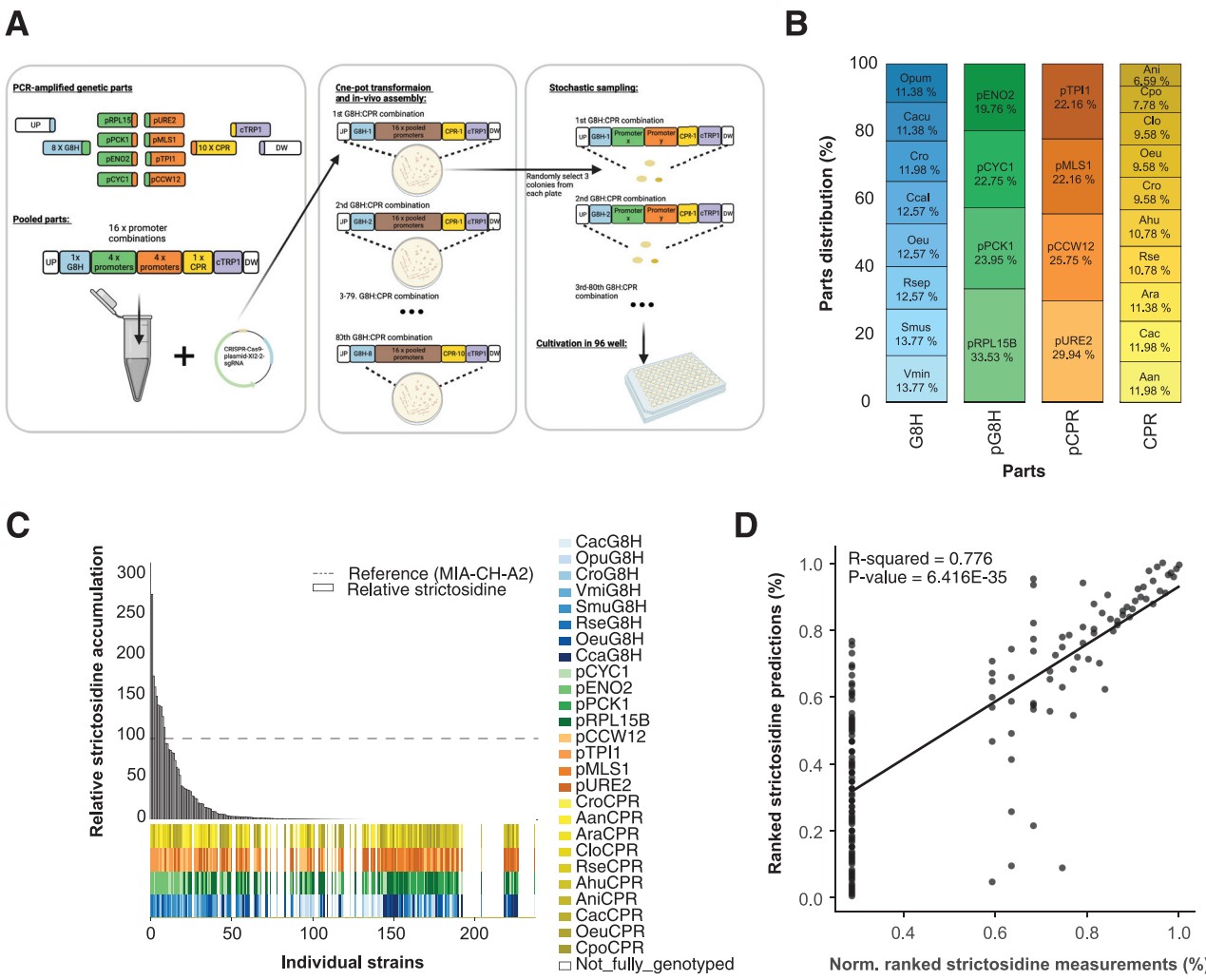

**Fig 3. Design, characterization and modeling of design-build-test-learn cycle I.** (**A**) Outline of the stochastic sampling and test workflow for data generation. Created with Biorender.com. (**B**) The distribution and counts of parts from the 167 strains that were accepted as input for machine learning in the first learning phase of the first DBTL cycle. (**C**) The distribution of observed strictosidine titers relative to reference strain MIA-CH-A2. Below the bar plot the distribution of parts for each of the 238 analyzed strains is presented. (**D**) Cross-validated predictions vs average normalized strictosidine production. All values are ranked.

and the number of wet lab steps. With CasEMBLR this would be enabled by stable genomic integration of assembled expression clusters as defined in the DESIGN step, and by the direct transformation of parts rather than the adoption of shuttle vectors propagated via a bacterial host, respectively. For this purpose, CasEMBLR is a well-suited tool as it leverages high-fidelity multi-part assembly based on homologous recombination and limits expression stochasticity by genomic integration of the assembled expression units. The seven parts encode two different promoters each controlling the expression of a gene encoding a G8H or a CPR, together with a selectable marker and two homology regions for the genomic landing pad (Fig 3A).

When generating diversity, it is essential to remember that the outcome vs effort is restricted in the build and test part of the DBTL cycle due to physical capacities in strain construction and testing. A potential way to accelerate the process is with stochastic variant generation [37]. Hence, we used teemi to output parts lists for combinatorial assemblies, each encoding a single G8H:CPR combination together with all 16 different promoter

combinations (4x4)(05_1_BUILD_Combinatorial_library)(Fig 3A). The designs were assembled as one-pot transformations together with the selectable marker and the two up- and down-homology regions, making each transformation consist of 21 parts for a total of 16 genetic designs in each of 80 (8 x 10) transformations (05_1_BUILD_Combinatorial_library, **Paragraph: 4**)(Fig 3A).

For the TEST module, we scored genotype and phenotype relationships of stochastically sampled colonies based on DNA sequencing of promoter:gene combinations and liquid-chromatography mass spectrometry to quantify strictosidine, respectively (Fig 3A). The choice on which data to capture in the TEST module was guided by simple principles. First, working on engineering biology, bioengineers are often aiming to correlate, and even predict, a sought-for phenotype of a living system based on a given genotype. While genotype is largely described by DNA sequencing, the phenotype of interest can be assessed by a multitude of parameters and proxies, such as fitness, shape, taste, and function. In this study, the objective function was to adopt teemi for the cost-effective engineering of cell factories and the prediction of production titers in engineered cell factories based on their respective genotypes. As such, the minimal set of input data to be used for model-guided optimization of cell factories would be genotype and titers based on Sanger-based DNA sequencing and quantitative liquid chromatography-mass spectrometry data, respectively.

Sequencing results matched with the simulation from the DESIGN step, were organized into a CSV file containing rows with strains that had unambiguous genotypes along with strictosidine production as part of the teemi workflow (06_1_TEST_Library_characterisation, **Paragraph: 3**). The pooled library approach complemented stochastic sampling of three colonies from each transformation to maximize diversity generation within the shortest amount of time [37], from which 159 unique genotypes were extracted from the 238 sampled colonies (12.4% coverage of the 1,280 design solution space)(06_1_TEST_Library_characterisation, **Paragraph: 4**). Furthermore, out of the total 159 unique genotypes obtained, the distribution of the 8 different G8Hs and 10 different CPRs were 11.4–13.8% and 7–12%, respectively, while for the 4 promoters driving expression of genes encoding G8Hs and CPRs the distributions were 19.6–33.5% and 22.2–30.0%, respectively (06_1_TEST_Library_characterisation, **Paragraph: 4**)(Fig 3B), totaling a deviation span of 1.4–8.5 percentile points from an even distribution. Taken together, these results demonstrate efficient parts assembly and relatively large coverage of the theoretical sequence space. With respect to strictosidine production, LC-MS measurements were obtained concomitantly, and data was normalized by the mean of the production obtained for the reference strain MIA-CH-A2 run in technical quadruplicates on three different replicate plates (29.29 +/- 4.84 μM; 34.77 +/- 4.85 μM; 34.23 +/- 7.60 μM) [29] (06_1_TEST_Library_characterisation, **Paragraph: 2.1**); 10_2_TEST_Library_characterization, **Paragraph: 1.3**). From the analysis, 9 of the 238 strains tested were observed to produce more than the reference strain (Fig 3C and S2 Table).

Lastly, in the interest to automate the modeling of genotype and phenotype data and to recommend forward-engineering of lead strains beyond those already used for modeling, we showcase integration of machine learning by teemi to LEARN genotype-phenotype relationships as well as recommend new strain designs not seen in the training data set. Commonly used machine learning models include XGBoost models which have previously been shown to have high predictive capabilities in modeling gene expression values [38], as well as genetic and metabolic networks [39]. Complementary to this, DeepLearning algorithms have been used in a plethora of bioengineering applications notably to predict guide-RNA activity for CRISPR/Cas-based genome engineering [40] and gene regulation [41]. Additionally, stacked ensembles, distributed random forest (DRF), general linear models (GLMs), and gradient booster models (GBM) models have also found applications in various biological areas [42,43].

**Table 3. Machine-learning model characteristics.**

| Model | First model | Second Model |
|---|---|---|
| | Deep Learning | XGBoost |
| MAE* | 2.728627138605333 | 8.669277115850836 |
| RMSE* | 6.088407587552942 | 19.04539155210566 |
| Cross-validation MAE** | 8.037346736078618 | 11.928834673923415 |
| Cross-validation RMSE** | 18.104192131285426 | 23.340093018693615 |
| $R^2$ of observed vs. cross-validation-predicted | 0.776 | 0.850 |

*Reported on train data

**Reported on cross-validation data

Taken together, the adoption of such a vast repertoire of computational methodologies emphasizes the diverse degree of complexity found in nature and biological research. However, as no single machine learning algorithm is optimal for all learning tasks [44,45], 1,895 different models, including DRF, GLM, XGBoost, GBM, DeepLearning, and StackedEnsemble sourced from H2O AutoML [46], were made as a function of gene and promoter combinations combined with normalized strictosidine measurements (input_for_ml_dbtl1.csv). AutoML was used to investigate the performance of all models with different algorithms and different hyperparameters instead of manually changing parameters of different models, or even performing manual pattern investigation. The simultaneous investigation of all the different models using this approach facilitated the training of different models on the single regression learning task of predicting the ability of the design combinations of G8H and CPR expression cassettes to increase the strictosidine production, and indirectly its ability to transform geraniol to 8-hydroxygeraniol. Through a 1-hour run of AutoML in H2O, a deep learning model was found to be the best-suited model for predicting relative production from genotypes (Fig 3D). The best model was found by sorting on MAE from cross-validation data with the best-performing model yielding an MAE = 8.03 and an RMSE = 18.10 based on 10-fold cross-validation predictions (07_1_LEARN_Modelling_and_predictions, **Paragraph: 2**)(Table 3). These MAE and RMSE values represent ~ 3 and 7% of the full range of measurements (0.0–245.0), respectively.

The strictosidine measurements were transformed into ranked values representing 245.03 = 1.00, 156.32 = 0.99,. . ., and 0.00 = 0.18 of the full range of measurements, respectively (Fig 3D). Observed production values of 167 strains were compared to cross-validation predictions, with the deep learning model yielding an overall $R^2$ = 0.77 (Fig 3D). However, the model tended to underpredict production, as evidenced by the majority of predicted strictosidine levels lying below the observed production curve (Fig 3D).

Beyond the motivation to model genotype-phenotype landscapes from genotypes and strictosidine production profiles for the 167 strains used for model training, a further motivation was to use the deep learning model to explore genotypes not seen in the training data set. From the remaining 1,121 theoretical combinations, 42 genotypes were predicted to produce more strictosidine than the reference strain (3.84% of the uncharted theoretical design space). With a fully deployed DBTL workflow now available in teemi, we were thus motivated to efficiently explore the combinatorial design space via a second DBTL cycle.

## teemi for design-build-test-learn cycle II

From the learnings of the first DBTL cycle, we used teemi to design the next DBTL cycle using the parts found in experimentally-validated top-performers with previously non-observed

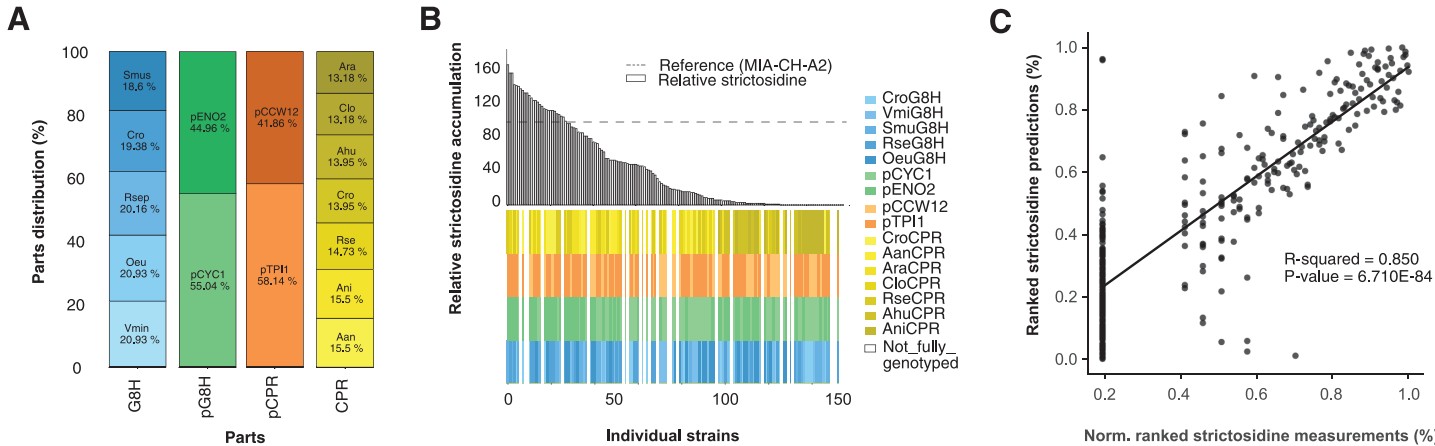

**Fig 4. Design, characterization and modeling of design-build-test-learn cycle II.** (**A**) The distribution and counts of parts from the strains that were accepted as input for machine learning in the second learning phase of the second cycle of DBTL. (**B**) The distribution of observed strictosidine titers relative to reference strain MIA-CH-A2. Below the bar plot the distribution of parts for each of the 240 analysed strains is presented. (**C**) Cross-validated predictions vs average normalized strictosidine production. All values are ranked.

combinations from the machine learning-guided predictions of the first DBTL cycle (08_2_DESIGN_Model_recommended_combinatiorial_library, **Paragraph: 1**). Balancing the maximum remaining search space (1,121 designs) and the predictive power of the deep learning model trained on data from the first DBTL cycle (159 designs), we decided for a maximum build capacity of 180 strain designs based on the parts found among the 20 predicted top-performers. This resulted in a distribution of 5 G8Hs, 2 promoters for controlling expression of genes encoding G8Hs, 2 promoters for controlling expression of genes encoding CPRs, and 7 CPRs (16 parts in total, creating a theoretical combinatorial space of 140 strains)(08_2_DESIGN_Model_recommended_combinatiorial_library, **Paragraph: 1**)(Fig 4A). Based on these parts, the combinatorial optimization approach was conducted as in the first DBTL cycle (Fig 3A). Of note, by choosing this combinatorial optimization approach we were biased towards "generating more hits-on-target" rather than the exactness of the constructed genotypes. This said, among the top-20 designs investigated, that form the basis of our sublibrary in the second DBL cycle, 16 of the highest predicted designs were made (16/20 = 80%), indicating only a modest trade-off between exploration and exploitation within the constrained design space of this study.

In the BUILD step, the combinatorial optimization resulted in 35 transformations of pooled transformations, this time consisting of a background strain and 11 different parts, namely 2 G8Hs (1 G8H x 2 overhangs), 2 promoters for expression of G8H-encoding genes, 2 promoters for expression of CPR-encoding genes, 2 CPRs (1 CPR x 2 overhangs), a TRP1 expression cassette, and UP and DW homology regions. This created a sequence space of 140 (35 x 4) unique 9 kb 7-parts assemblies at the target genomic locus. From the transformations, 4 strains with known G8H and CPR were sampled randomly from each plate to get 140 strains, a number that matches the sequence space (2 extra were sampled from one strain, therefore 142 in total). Additionally, 2 blanks, 2 negative controls and 22 positive controls were sampled totaling 168 strains (09_2_BUILD_Combinatorial_library, **Paragraph: 8**) (Fig 4).

For the TEST step, genotypes and strictosidine production levels were again assessed by DNA sequencing and LC-MS, respectively (Fig 4A and 4B). From sequencing, a total of 86 unique genotypes were obtained from the 142 colonies sampled (86/142 = 60.56% coverage), of which 75 were not present in the first round, while the number of duplicates was 43 (10_2_

TEST_Library_characterization, **Paragraph: 5.2**). Out of the 86 unique genotypes, the distribution of the 5 different G8Hs and 7 different CPRs were 18.6–20.9% and 13.1–15.5%, respectively, while for the 2 promoters each driving expression of genes encoding G8Hs or CPRs the distributions were 55.0–45.0% and 58.1–41.9%, respectively (Fig 4A, 10_2_TEST_Library_characterization, **Paragraph: 4**). From the first cycle, the 159 unique strains were generated in 80 transformations, providing 99% more strains compared to what could maximally be obtained from single-design transformations. The second cycle generated 86 unique strains in the 35 transformations and generated 145% more strains than single-design transformations. Combined with the 159 unique strains generated in the first cycle, there were 234 (159+75) unique genotypes created from a total of 115 transformations, with only 62 identical genotypes harvested in both cycles, highlighting once again stochastic sampling from pooled transformations as an efficient approach for searching amble genotypic spaces.

Again, and concomitant to sequencing, the strictosidine titers were measured for all 142 strains as well as replicates of reference strain and positive controls (with known production) and negative controls (no G8H and CPR expression cassette inserts, 168 in total). From the 142 forward engineered strains, 28 strains produced more strictosidine than the reference MIA-CH-A2 strain (28.11 μM), which is a 5-fold improvement in performance compared to the first DBTL round ((28/142)/(9/238) = 5.21), and the highest producing design being pENO2:*Smus*G8H and pTPI1:*Rse*CPR with 69% higher production(47.69 μM) compared to the reference strain (Fig 4B and S3 Table, (10_2_TEST_Library_characterization, **Paragraph: 1.3**)). 13 strains were only partially genotyped and were therefore discarded. Combining strictosidine measurements with genotyping resulted in 129 accepted strains, of which 86 were unique (86/129 = 66.67% of the theoretical sequence space)(Fig 4B, Accepted strains in second iteration, 10_2_TEST_Library_characterization).

As conducted for the first DBTL cycle, we used AutoML to investigate and rank the performance of 774 models. As AutoML trains models until they reach convergence, the number of models is only limited by the time which is set dynamically by H2O to 1 hour if the number of models is set to "None" (H2O documentation) [46]. Through a 1-hour run of AutoML in H2O, a XGBoost model was found to be the best-suited model for predicting relative production from genotypes. Similar to the first DBTL cycle, the best model was found by sorting on MAE on cross-validation data with the best-performing model yielding an MAE = 11.93 and an RMSE = 23.34 based on 10-fold cross-validation predictions (11_2_LEARN_Modelling_and_predictions, **Paragraph: 4**). These MAE and RMSE values represent ~ 7 and 14% of the full range of measurements (0.0–170.0), respectively. Additionally, the best model had an overall correlation coefficient of $R^2$ = 0.85 when ranking observed production titers with cross-validated predicted titers of the 296 strains (Fig 4C).

Furthermore, and as exemplified in the first DBTL cycle (Fig 3C and 3D), the LEARN step focused on parts distribution and correlation coefficient between the ranking of observed vs. cross-validated predicted strictosidine titers to inform about the possible impact of using the models generated from data in the first DBTL cycle for a second DBTL cycle. Here, when asking the best-performing XGBoost model trained on the data generated in the second DBTL cycle to recommend parts to be used for forward-engineering of new strains with high(er) even strictosidine titers in a potential third DBTL cycle, we found that the Top-25 predictions overlapped by 70.0% with those already exploited for the second DBTL cycle (S3 Fig, (11_2_LEARN_Modelling_and_predictions, **Paragraph: 5.3.1**)). Furthermore, while the hit-rate of high-producers compared to the reference strain obtained in the second DBTL cycle increased compared to the results from the first DBTL cycle (9/238 = 3.8% vs 28/142 = 19.7%), and a modest increase in correlation coefficient between the ranking of observed strictosidine and

predicted production could be obtained ($R^2$ = 0.77 vs. $R^2$ = 0.85)(Figs 3D and 4C), although the algorithm was not able to precisely rank the genotypes according to production (S4 Fig).

## Stop-go evaluation for design-build-test-learn cycle III

To further evaluate whether to continue into a third DBTL cycle in search for high(est) producing strain designs, we used several assessment criteria. First, we evaluated the coverage of the explored design space across the two engineering cycles, totaling 234 different designs out of 1,280 possible combinations (18.3%). Second, we used learning curves as a quantitative parameter to guide the stop-go evaluation (07_1_LEARN_Modelling_and_predictions, **Paragraph: 8**); 11_2_LEARN_Modelling_and_predictions, **Paragraph: 6**)). Learning curves, created based on the MAE compared to the number of data points used for training, can give an indication of how adding more data could affect the predictive power of the models used between iterative DBTL cycles.

We can use data partitioning to evaluate how well a model performs and behaves with different subsets of the data [47]. To do this, we shuffled the data, divided it into parts, and trained a model on each part of the dataset. We repeated this process 10 times (07_1_LEARN_Modelling_and_predictions, **Paragraph: 8**); 11_2_LEARN_Modelling_and_predictions, **Paragraph: 6**). When comparing the learning curves obtained from data generated in the first DBTL cycle vs. the second DBTL cycle, it can be observed that the MAE of the cross-validation decrease through the data points but with a reducing slope as data used for training increases (slope -0.009x from datapoint 56–167 for the first DBTL cycle and slope -0.016x from datapoint 60–296 for the second DBTL cycle)(Fig 5A and 5B). For DBTL cycle I, the lowest MAE from the training data obtained was 0.08 and with a decreasing trend even when the models were trained on 167 data points (07_1_LEARN_Modelling_and_predictions). For the learning curve obtained from the cross-validated models trained on data from DBTL cycle II, the training MAE, on the other hand, reached a plateau at 120 data points with a minimum MAE of 2.56 (Fig 5B), indicating that the model does not improve much with more data, even though the correlation coefficient (Fig 4C), and thereby predictive power, increases in DBTL cycle II (Fig 5B, 11_2_LEARN_Modelling_and_predictions).

Lastly, and in addition to the 70% overlap between the Top-25 predictions offered by the best-performing XGBoost model and the designs already exploited for the second DBTL cycle (S3 Fig), we also compared the distribution of best-performing observed strictosidine producers arising from the first and second DBTL cycle. Here we observed that even though the best-performing strain design compared to the reference strain was identified in the second DBTL cycle, the increase in production compared to the best-performing strain observed from the first DBTL cycle was merely 10% (159 vs 144)(Fig 5C).

Taken together, evaluating design space coverage, learning curves, and observed production from top-ranking designs between individual DBTL cycles, can help guide decisions as to whether to stop further exploration of the remaining design space or to continue forward engineering in search of the global maximum. In this case, the relative high design space coverage (18.3%), the stagnating learning curve and the higher MAE in the second DBTL cycle, and the overlap between already-explored designs and the forward engineering predictions offered by XGBoost for a third DBTL cycle, supported a "stop" decision on further exploration of this design space.

## General design highlights

With the finalization of the two iterative DBTL cycles for the multivariate optimization of the C8-hydroxylation of geraniol to 8-hydroxy-geraniol, several design take-homes can be

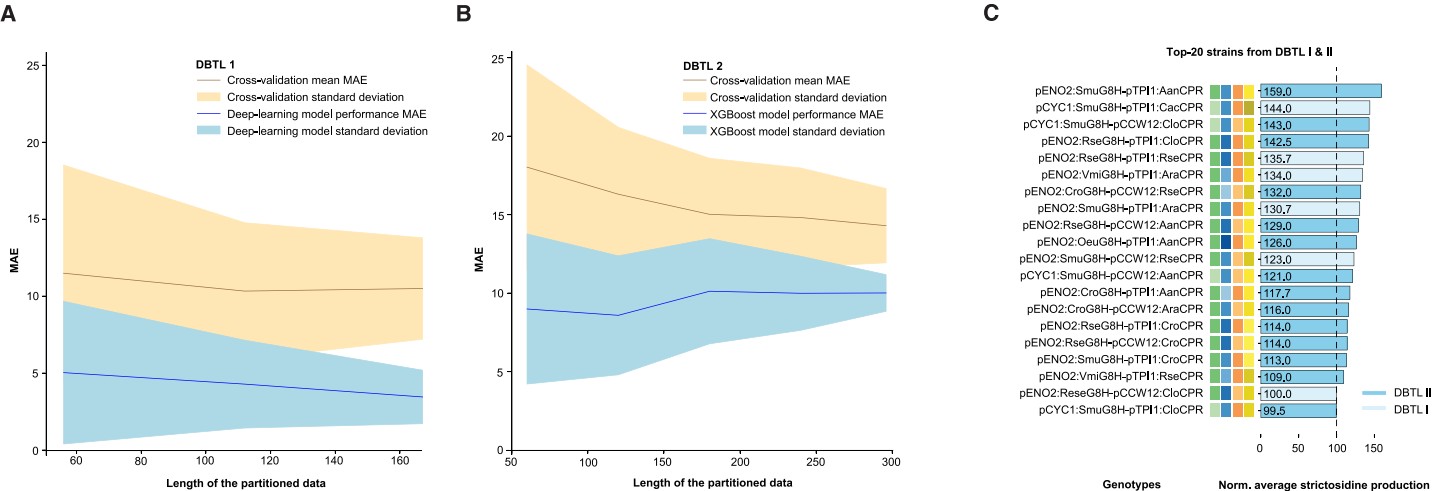

**Fig 5. Learning curves and top-ranking strains designs from the iterative engineering cycles.** Learning curves from the first (**A**) and second (**B**) DBTL cycles, illustrating mean absolute error (MAE) of the best-performing deep learning and XGBoost models used cycle I and II, respectively, in relation to the number of data points (blue line) and the cross-validation holdout prediction MAE together with the standard deviations of the 10 models created (yellow line). The points are based on 10 models created with a randomized shuffled data in partitions of 33, 67, 100% and 20, 40, 60, 80 and 100% of the data available for dbtl1 and dbtl2 respectively to get the same size of partitions. (**C**) Average strictosidine production for Top-20 strains from first and second DBTL cycles. Genotypes are shown (left) with their respective color codes (middle) and average strictosidine production (right). For the strictosidine production, the light and dark blue colors correspond to strain designs that were first found in the first and second second DBTL cycle, respectively.

extrapolated. Firstly, we found that the strictosidine production increased by up to 59% comparing the reference strain (MIA-CH-A2) with the best-producer encoding pENO2:*Smus*G8H and pTPI1:*Aan*CPR (28.115095 μM vs. 44.719792 μM, respectively) (Fig 5C). Next, from the top-ranking strictosidine producers, our results indicate a high level of CPR promiscuity for the two top-ranking G8H candidates from *Swertia mussatii* (*Sm*u) and *Rauvolfia serpentina* (*Rse*), as evidenced by 5 different CPRs included in the Top-6 ranking strain designs (Fig 5C). Notably, the identification of several CPRs improving 8-hydroxygeraniol synthesis corroborates previous findings [27]. Furthermore, even though the G8H from *Catharanthus roseus* (*Cro*) has been critically acclaimed to enable high production of 8-hydroxy-geraniol and down-stream plant bioactives [26,27,29], this study highlights *Smu*G8H and *Rse*G8H as promising geraniol hydroxylating enzymes in microbial cells (Fig 5C). Lastly, and interestingly, the promoters driving the expression of genes encoding CPRs, promoters with high expression during the glucose-rich early- and mid-exponential phases of cultivation, such as pTPI1 and pCCW12 are prominent design parts (Figs 2B, 2C, and 5C). For G8H expression control, top-ranking strain designs also included strong promoters, albeit promoters with expression lower than pTDH3, used in the reference design of strain MIA-CH-A2 (Fig 5C), thus indicating that use of strong constitutive promoters may be dispensable for P450-mediated biocatalysis in yeast.

Taken together, the multivariate design space explored and exploited in this teemi testbed has offered robust take-homes in terms of bioengineering designs benchmarking with, and extending beyond, previously reported G8H and CPR studies.

## Discussion

The aim of this study was to showcase teemi for bioengineering demonstrated experimentally via a multi-variate biological testbed founded on i) computer-aided design to standardize

workflows and minimize errors during the build step, ii) stochastic sampling from pooled DNA parts libraries, iii) research data management according to FAIR principles, and iv) the use of 2,000+ ML models sourced from AutoML to stress-test predictive engineering compared to manual extrapolation of patterns.

The iterative bioengineering testbed supported by teemi, not only enabled a streamlined workflow for quantitative assessment of genotypes and phenotypes, but also supported objective decision-making. For instance, the best models showed good correlations for both first and second DBTL cycle (MAE = 2.7% and 8.7% of the measurement ranges, respectively (Table 3), with an increase in predictive power from the first DBTL to the second ($R^2$ = 0.776 and 0.850, respectively)(Figs 3D and 4C), ultimately increasing the forward-engineering hit rate (i.e. obtaining phenotypes that performed better than the reference strain within the sequence space) by more than 5 times (from 3.8% to 19.7%)(S4 Fig). Also, we observed that as we generated more data, the cross-validated MAE decreased in both first and second DBTL cycle (slopes of -0.009x and -0.016x, respectively)(Fig 5A and 5B). Having said this, for this particular testbed, we observed different trends in the two learning curves regarding the test MAE, where the models in the first DBTL cycle seem to overfit the data, and the models in the second DBTL cycle seem to converge, while from cross-validation we observed a higher MAE for the models in the second cycle compared to the first cycle (MAE = 8.04 and 11.93, respectively)(Table 3). The higher MAE for the second cycle is likely caused by higher variation in the data points between the different test runs, and call for higher quality analytical data. Furthermore, as almost 20% of the design space has been explored during the first two DBTL cycles, and with a mere 10% improvement in strictosidine production in top-ranking design found in the second DBTL cycle compared to the top-ranking hit from the first DBTL cycle, a natural next engineering step would be to focus attention to other limiting factors of the strictosidine pathway, such as rational engineering of the other hydroxylation steps [26], for instance using the design principles uncovered in the best–performing G8H:CPR step.

To generate top-performing strains requires high-performing genetic parts and finding top-performing genetic parts is a hurdle when constructing biosynthetic pathways. Here we used BLAST to find homologs to catalyze our challenging hydroxylation step, but an alternative approach would be to BLAST and cluster sequences with, for example, the open source tool MMseqs2 [48] and introduce variants from each cluster to increase library diversity. Another way to find alternative genetic parts, circumvent bottlenecks in the pathway, or find novel biosynthesis routes would be to use retrobiosynthesis tools like Retropath 2.0 [49] or the BNICE framework [50]. These strategies align with the iterative nature of the DBTL cycle, where each step is crucial for efficient pathway engineering.

As in any DBTL cycle, the goal has focused on maximizing the knowledge generated, and ultimately reducing time and resource allocation during iterative bioengineering cycles. With the step-by-step guidance illustrated by experimental data in this study, we expect that the use of FAIR-compliant teemi will enable i) that more experiments can be performed in shorter amount of time and with less errors, ii) better integration of IT tools with other resources (e.g. human-centered and/or robotic work-flows), and iii) effective inter- and intra-laboratory knowledge transfer, and thus drastically increase reproducibility and standardization in biology. With respect to better integration of IT tools in bioengineering, it deserves to be mentioned that this study was co-led by MSc-level students to maximise compatibility with both the skills and the aspirations of early-stage bioengineers. Basic programming skills are advantageous in order to benefit from all the capabilities of teemi, but not needed to get started. Indeed, in teemi, abstractions are used to streamline workflows and manage complexity, and by providing these workflows as open-source for the community, we can continuously improve the workflows and learn a lot more from each other and in less time.

## Materials and methods

### Executing teemi

teemi is distributed as free open-source software at pypi.org. To maximize the usefulness of teemi we have developed a set of Jupyter notebooks that can be executed locally or through Google Colaboratory without any prior installation of software. This, we believe will lower the time spent on installation and resolving dependencies which is useful for all users regardless of programming experience. The only requirement is a Google account to use Google Colaboratory and the notebooks can be found at https://github.com/hiyama341/teemi/tree/main/colab_notebooks.

### Modules of teemi

teemi consists of four modules that aid in strain construction through the Design, Build, Test, and Learn phase of the DBTL cycle with an additional Laboratory information system module (LIMS)(Fig 1). The first module is DESIGN, which includes functions for cloning procedures, and the generation of combinatorial libraries. The second module, BUILD, is focused on building strains with functions for simulating and calculating PCRs, transformation reactions and automatically generating robot executable instructions. The third module is the LIMS module that can import and export DNA sequences and keep track of samples through Benchlings API and a local CSV file database. The fourth module, the TEST module, has functions to pre-process data from sequencing results and infer the relationship between sequencing results and genetic parts based on pairwise alignment. The final module is aimed at the LEARN phase by incorporating easy-to-use ML functions with plotting functions.

As teemi is under MIT license anyone can edit, and use the code rendering it flexible and reusable. Additional modules can be added to the package by anyone willing to contribute or modification of the code by the users is allowed. The guidelines for contributing can be found on teemi's contributing site here. For a high-level overview of teemi, please visit teemi's documentation page https://teemi.readthedocs.io/en/latest/. The site provides detailed descriptions of the modules, functions, and classes and how to install teemi locally.

### teemi: Simulation of experimental workflow and data analysis

To enable reproducible high-throughput strain construction, literate programming along with the modules of teemi was used to simulate all experimental workflows used in this study. The experimental workflows were divided into Jupyter notebooks encompassing different parts of the DBTL cycle as shown in Table 2.

This framework enabled the generation of a large number of strains while keeping mistakes at a minimum by simulating experimental workflow and keeping track of samples. More specifically, It provided a tool to simulate the amplification of DNA in PCR reactions, retrieving locations of all relevant DNA fragments and primers while attaining an overview of the procedures. PCR and transformation mixes were calculated and simulated *in silico*. Additionally, it worked as a laboratory notebook containing all experimental setups, observations, and results. These notebooks also show how teemi and literate programming can incorporate advanced machine learning models through H2O's AutoML package (07_1_LEARN_Modelling_and_predictions, 11_2_LEARN_Modelling_and_predictions).

### Experimental strains used in this study

The *S. cerevisiae* strains constructed in this study were derived from the MIA-CH-A2 strain containing CroG8H, CroCPR, and 11 other genes under promoters pTDH3 and pTEF1,

driving the biosynthesis pathway from geraniol to strictosidine [29]. The background strain used in this work was made by using literate programming along with teemi's design and build modules to enable CRISPR-mediated knockout of *Cro*G8H and *Cro*CPR in the Easy-Clone site X-3 and XI-3 sites [36], respectively. These modules made it possible to extract the knockout sites and simulate the *in vivo* assembly while generating GenBank files of the newly generated strains (04_1_BUILD_Background_strain). The resulting background strain was named MIA-HA-1 (MIA-HA-1.gb).

## Microbial strain cultivations

We used teemi and literate programming to document and calculate all steps of plate and liquid cultivations. The plate and liquid cultivations were performed as described in [29] except that 0.2 mM geraniol and 1 mM tryptamine were added to YPD media and the cultures were grown at 300 rpm when testing for strictosidine production (03_1_BUILD_gRNA_plasmid, **Paragraph: 1.5**, 04_1_BUILD_Background_strain,, **Paragraph: 6**, 05_1_BUILD_Combinatorial_library,, **Paragraph: 4**).

## Genetic parts selection

To standardize the selection of genetic parts we developed an algorithm in a literate programming workflow that automates the selection process by searching and selecting homologs based on amino acid identity through NCBIs databases (00_1_DESIGN_Homologs). Using *Catharanthus roseus* sequences (Q8VWZ7, Q05001) as queries, eight G8H and CPR genes were found on NCBI's databases. To diversify the CPR genes we searched the PhytoMetaSyn database using *Catharanthus roseus* CPR mRNA (X69791.1) as a query. We selected two additional CPRs from the largest ORFs of the mRNA transcripts, which provided a broad range of amino acid identities for all the chosen CPRs (00_1_DESIGN_Homologs, **Paragraph: 5.1.5**). The sequences were codon-optimized for S.cerevisiae with DNA Chisel (00_1_DESIGN_Homologs, **Paragraph: 5.3**).

A literate programming workflow was used to select promoters to drive the expression of the gene homologs (01_1_DESIGN_Promoters). Promoters were chosen based on absolute mRNA abundance measured from S. cerevisiae CEN.PK 113-7D at cultivation time points 6, 12, and 24 hours (01_1_DESIGN_Promoters, **Paragraph: 2**) [34].The promoters were defined as 1kb upstream of the target gene, with lengths varying from 984–1004 bp due to differences in in our in-house strains and the database strains. Four promoters were selected for each *CYP* and *CPR* module based on constitutive expression and expression patterns (high/low and increasing/decreasing). To prevent homologous recombination during transformation, all promoter sequences were aligned to ensure no homologous sequences, reducing the chance of genetic parts looping out (01_1_DESIGN_Promoters, **Paragraph: 8**). To streamline the combinatorial library size and minimize the number of integrated fragments, gene homologs were assembled with tCYC and tADH terminators.

## Extracting genetic parts

In this study, we ordered gene homologs as gBlocks and cloned them into plasmids along with tADH and tCYC terminators. To extract promoters from the genomic DNA of wild-type *S. cerevisiae* CEN.PK2-1C, we generated specific primers. Additionally, we amplified the TRP1 cassette from plasmid pRS414-USER using primers overlapping with the tCYC1 terminator and homologous to the fragment downstream of EasyClone site XI-2 [36].

Prior to conducting the PCR reactions and USER assemblies in the laboratory, we simulated them using teemi's module PCR.py. The specific PCR programs, polymerases,

purifications methods, and amplification of USER and transformation parts can be found in (03_1_BUILD_gRNA_plasmid, 04_1_BUILD_Background_strain, 05_1_BUILD_ Combinatorial_library).

## Plasmid construction

Construction of plasmids was simulated with teemi's cloning.py module in a literate programming workflow and assembled in the lab with USER cloning [51], 03_1_BUILD_gRNA_ plasmid). More specifically, the cloning.py module was used to simulate and construct the plasmid (Double_gRNA_vecor_p1_G09_(pESC-LEU-gRNA_ATF1_CroCPR) that was used to perform CRISPR-mediated deletion of the G8H and CPR genes (03_1_BUILD_gRNA_ plasmid). The second plasmid used in this study, pESC-URA-gRNA_XI2-2, used for in vivo assembly into the EasyClone site XI-2 locus had been constructed previous to this work.

## Designing genetic parts for the combinatorial library

We used two methods to design combinatorial libraries, the commercially available Teselagen Design module, and our own open-source DesignAssembly algorithm. The designs made with the DesignAssembly algorithm incorporate 40 bp overlapping overhangs by default with a distribution of 50/50% of the overhang to the forward and reverse primer. A pad (defined as a nucleotide sequence of 40 bp) was incorporated between the promoters with an *ATF1* gRNA site to provide the deletion of the module at a later stage. The designs and instructions for the assembly can be found in the following notebook (02_1_DESIGN_Combinatorial_library).

Another similar combinatorial library was created with Teselagen Design Module software where the parts were made with 30 bp overhangs. Annealing temperatures were re-calculated with tmcalculator.neb.com. The design of overhangs can be seen here (05_1_BUILD_ Combinatorial_library). Both designs are presented in this work but it was decided only to go forward with the designs made with Tesselagen (05_1_BUILD_Combinatorial_library).

## Pooled construction of the combinatorial libraries

The combinatorial library in this study was constructed using the CasEMBLR method and designed for the EasyClone site XI-2 [36]. To facilitate the construction process, we used literate programming and teemi's modules to standardize and simplify the procedure (05_1_ BUILD_Combinatorial_library, 09_2_BUILD_Combinatorial_library). We used the teemi's lab module (PCR.py and transformation.py) to calculate PCR melting temperatures, simulate and verify gel bands, and track samples using a local CSV-based LIMS system (csv_database. py,). The plasmid pESC-URA-gRNA_XI2-2 was used for the in vivo assembly of the library into locus XI2-2. Flanking regions for repair were approximately 0.5 kb, and the homology regions between parts were 30 bp by default. A tryptophan selection marker was used to select for positive transformants. (05_1_BUILD_Combinatorial_library, 09_2_BUILD_ Combinatorial_library).

To create the library, we pooled genetic parts into one mixture, including promoters, UP, DW, and cTRP1 parts, and one gene pair, all with overlapping overhangs and in equimolar amounts(05_1_BUILD_Combinatorial_library, **Paragraph: 2.2**), 09_2_BUILD_ Combinatorial_library, **Paragraph: 5.1–5.3**). The pooled library was transformed with the genetic parts in a one-pot reaction to prevent unwanted homologous recombination between the genes (05_1_BUILD_Combinatorial_library, **Paragraph: 4.1**, 09_2_BUILD_ Combinatorial_library, **Paragraph 6.1–6.3**)).

Yeast transformations were carried out using the LiAc/SS carrier DNA/PEG method [52] and performed with 1–2 ml of a background strain with an optical density of 1. Each

transformation reaction contained 0.25 pmol of a CRISPR plasmid expressing the gRNA for XI-2 and 0.5 picomoles of each DNA fragment (05_1_BUILD_Combinatorial_library, **Paragraph: 4**, 09_2_BUILD_Combinatorial_library, **Paragraph: 6.1**).

Control strains were transformed alongside the library strains. These strains were transformed with plasmids containing uracil or tryptophan to test transformation efficiency and water to test cell viability. The first set of transformations was split into three in the first round of the DBTL(05_1_BUILD_Combinatorial_library, **Paragraph: 4.1**, while the second set was split into two in the second cycle (09_2_BUILD_Combinatorial_library, **Paragraph: 6.1**).

## Sample preparation for LC-MS and data analysis

Sample preparation and internal standards were prepared according to [29], with the exception that pre-cultures were transferred to media containing 0.2 mM geraniol + 1 mM tryptamine after two days as described in 05_1_BUILD_Combinatorial_library**(Paragraph: 5)** and 09_2_BUILD_Combinatorial_library**(Paragraph: 8)**. The metabolites strictosidine, loganic acid, loganin, secologanin, and tryptamine were analyzed according to [29].

The full data analysis with respect to normalization, and calculations can be found in notebook 06_1_TEST_Library_characterisation and 10_2_TEST_Library_characterization where functions from teemi's data_wrangling.py were used to process the data.

## Promoter genotyping

Genomic DNA was extracted from overnight cultures with LiOAc/SDS method adapted for 96 well plates [53]. Each extract was used as a template for two PCR's spanning the promoter gene pairs (05_1_BUILD_Combinatorial_library, **Paragraph: 5**, 09_2_BUILD_Combinatorial_library, **Paragraph: 7**), providing approximately ~2700 bp and ~3200 bp (Lenghts_of_constructs). The colony PCR products were validated with 1% agarose gels followed by sequencing. Positive colony PCRs were first sequenced by Eurofins, using a PlateSeq Kit for crude PCR products according to the manufacturer's instructions. Second re-sequencing was performed with previous transformants with 5 µl PCR products and 2 µl ExoSAP-IT enzymes (Thermo Fisher Scientific Inc.) heated to 37 ˚C for 15 minutes followed by 80 ˚C for another 15 minutes.

The sequencing data consisted of a plate report describing each well's average quality and sequencing files (.ab1). Using teemi's data_wrangling.py we automated data processing by filtering out low-quality alignments (average quality < 50, length used > 25). Then, using functions from genotyping.py we inferred the promoter relationship to the samples. Wells with multiple inferred promoters were filtered out. The final result was CSV files with inferred promoters for each well. These results were merged with LC-MS data, resulting in a CSV file with genotypes and normalized strictosidine production for the strains.

## AutoML and learning curves

In this study, we used the AutoML H2O python library version 3.38.0.4 to automate the machine learning process (AutoML H2O). The H2OAutoML class was initiated with an input dataframe (input_for_ml_dbtl1.csv, input_for_ml_dbtl2.csv), response column(norm.strictosidine) and specified feature columns(promoter:gene combinations). The feature columns were made categorical, and 10-fold cross-validation was performed. The trained models were saved in a leaderboard, and the best model was selected to predict phenotypes of unseen genotypes in the remaining combinatorial library (07_1_LEARN_Modelling_and_predictions**(Paragraph: 1–7)**, 11_2_LEARN_Modelling_and_predictions**(Paragraph: 1–5)**).

To generate a learning curve, the teemi module auto_ml.py was used on the datasets (input_for_ml_dbtl1.csv, input_for_ml_dbtl2.csv). Here, the main function divides the dataset into partitions that progressively increase in size, and then trains models on each partition. The function outputs a dataframe containing the name of the top performing model(sorted by MAE), the mean-absolute error, and cross-validated values. This was done ten times for each dataset, including a shuffling step between each run (07_1_LEARN_Modelling_and_predictions(**Paragraph: 8**), 11_2_LEARN_Modelling_and_predictions, **Paragraph: 6**).

### Model based genetic part recommendations

The genetic parts for the second DBTL round were selected by iterating through all predictions of non-encountered combinations (08_2_DESIGN_Model_recommended_combinatiorial_library(**Paragraph: 1**)). Each new genetic part was saved and once the total number of combinations reached the maximum capacity the iteration stopped. The encountered genetic parts were then used in the following DBTL cycle to investigate the best-performing parts of the combinatorial library (09_2_BUILD_Combinatorial_library).

### Dependencies

S1 Table provides a list of dependencies required to run teemi's modules. Specifically, it describes the minimum dependencies needed, while the optional test dependencies can be installed through the setup.py file. The installation of these can be done with the following command: *pip install* teemi*[dev]*. For executing the 00_1_DESIGN_Homologs and 01_1_DESIGN_Promoters notebooks additional requirements need to be installed. These packages include InterMines Python API and Edinburgh Genome Foundry's DnaChisel. However, through the Google colab notebooks, these dependencies are installed automatically.

### Supporting information

**S1 Fig. Overview of all the functions and file formats used in teemi for the present study.** Created with Biorender.com.
(TIF)

**S2 Fig. Using literate programming to A) Simulate a gel with one line of code and B) Run the gel with the amplicons.**
(TIFF)

**S3 Fig. Schematic representations of A) top 25 designed strains from DBTL2 and B) top 25 predicted strains from the updated machine-learning model after DBTL2.**
(TIFF)

**S4 Fig. A) Showing the observed strictosidine production values vs. the the cross-validated values from the model in the (A) first DBTL cycle and B) second DBTL cycle with all accepted strains.**
(TIFF)

**S1 Table. Shows the dependencies for teemi divided into three categories: Minimal, Test, and Extra dependencies.**
(XLSX)

**S2 Table. Sorted strain performance of all fully genotyped strains in the first DBTL round.**
(XLSX)

**S3 Table. Sorted strain performance of all fully genotyped strains in the second DBTL round.**
(XLSX)

## Author Contributions

**Conceptualization:** Søren D. Petersen, Nikolaus Sonnenschein, Michael K. Jensen.

**Data curation:** Søren D. Petersen, Lucas Levassor, Christine M. Pedersen, Lea G. Hansen, Jie Zhang, Ahmad K. Haidar, Michael K. Jensen.

**Formal analysis:** Søren D. Petersen, Lucas Levassor, Christine M. Pedersen, Ahmad K. Haidar, Nikolaus Sonnenschein.

**Funding acquisition:** Jan Madsen, Rasmus J. N. Frandsen, Jay D. Keasling, Nikolaus Sonnenschein, Michael K. Jensen.

**Investigation:** Lucas Levassor, Lea G. Hansen, Nikolaus Sonnenschein, Michael K. Jensen.

**Methodology:** Søren D. Petersen, Lucas Levassor, Christine M. Pedersen, Jan Madsen, Ahmad K. Haidar, Michael K. Jensen.

**Project administration:** Rasmus J. N. Frandsen.

**Software:** Søren D. Petersen, Lucas Levassor, Nikolaus Sonnenschein.

**Supervision:** Lea G. Hansen, Jie Zhang, Rasmus J. N. Frandsen, Tilmann Weber, Michael K. Jensen.

**Visualization:** Christine M. Pedersen.

**Writing – original draft:** Søren D. Petersen, Lucas Levassor, Michael K. Jensen.

**Writing – review & editing:** Lucas Levassor, Jie Zhang, Jay D. Keasling, Tilmann Weber, Nikolaus Sonnenschein, Michael K. Jensen.

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
