## [Decision Letter · Decision Letter 0]

6 Nov 2023

Dear Dr Krogh Jensen,

Thank you very much for submitting your manuscript "Literate programming for iterative design-build-test-learn cycles in bioengineering" for consideration at PLOS Computational Biology.

As with all papers reviewed by the journal, your manuscript was reviewed by members of the editorial board and by several independent reviewers. In light of the reviews (below this email), we would like to invite the resubmission of a significantly-revised version that takes into account the reviewers' comments. In particular, please take care to address the detailed and constructive comments of Reviewer 2 in full.

We cannot make any decision about publication until we have seen the revised manuscript and your response to the reviewers' comments. Your revised manuscript is also likely to be sent to reviewers for further evaluation.

Sincerely,

Ruth E Baker

Section Editor

PLOS Computational Biology

Ruth Baker

Section Editor

PLOS Computational Biology

Reviewer's Responses to Questions

**Comments to the Authors:**

Reviewer #1: The manuscript describes computational and experimental work which has been executed to a high standard, is novel, interesting and makes a significant contribution to the biotechnology field. As a structural biologist, I am not expert enough in bioinformatics to make an authoritative judgement, but the paper is well-written, technically sound and the standard of presentation is very high. I thought that the introduction was rather brief and would possibly benefit from citing related work of others in this field. Apart from that I am happy to recommend publication of the manuscript in its present form.

Reviewer #2: The article introduces the Python Software teemi, designed to assist in the organization and guidance for strain construction. It supports various steps in the DBTL cycle while adhering to FAIR principles. The authors provide a case study that involves simulating guided and iterative laboratory workflows to optimize strictosidine production in yeast.

While this work may be of interest to the community, its current form appears to be more a collection of scripts or a Python package than a comprehensive platform for DBTL. As a result, it's unclear from the paper if the contribution is substantial enough for publication in PLOS Computational Biology. A relevant extra effort would be needed to design such a comprehensive platform. Here are some issues that would in anycase need to be addressed before publication.

Major Comments:

1. A workflow of the tool is missing. This omission makes it difficult to gauge the tool's generalizability. To be considered a platform, it should be evident what inputs are required from the user and what outputs the platform provides at each step of the DBTL. Does the user need to modify internal code to use the tool? If so, it resembles a collection of files that could serve as templates or inspiration rather than a platform.

2. The title suggests that the use of literate programming is a significant contribution to DBTL cycles in synthetic biology. However, literate programming is widely used by the synthetic biology community for iterative DBTL, especially through Jupyter notebooks. This creates confusion. The title is misleading (the actual name of the package in Github is "a python package designed to make high-throughput strain construction reproducible and FAIR). It is crucial for the authors to clarify the primary contributions of their work.

3. Table 1 is notably incomplete. Please update it with platforms that have already been released and published, addressing the DBTL cycle. For example, the Galaxy project is cited, but what about GalaxySynbioCad, which is a Galaxy-based platform specifically tailored to the DBTL pipeline? (https://www.nature.com/articles/s41467-022-32661-x)

4. The main assumptions, decisions, and methods adopted at each stage of the DBTL should be clearly explained (in connection with comment 1). For instance, in the BUILD module, the authors adopted CasEMBLR for CRISPR/Cas9-mediated assembly, and in the TEST stage, they used DMA Sequence and LC-MS...

5. In the LEARN stage, the authors claim that the library for the next iteration is based on the top-performing predicted models. However, it seems that no combinatorial experimental design is applied to enhance the exploration of solutions. This could introduce bias into the results, and this aspect is not adequately clarified in the text.

6. The tool's features and functionalities need to be specified. The term "engineering of complex biosystems" is too abstract. The features supported are clearly specified in the documentation of the Python package, but not in the paper.

**Have the authors made all data and (if applicable) computational code underlying the findings in their manuscript fully available?**

Reviewer #1: Yes

Reviewer #2: Yes

PLOS authors have the option to publish the peer review history of their article (what does this mean?). If published, this will include your full peer review and any attached files.

Reviewer #1: **Yes: **Jonathan B Cooper, UCL.

Reviewer #2: No
---

## [Decision Letter · Decision Letter 1]

17 Feb 2024

Dear Dr Krogh Jensen,

We are pleased to inform you that your manuscript 'teemi: An open-source literate programming approach for iterative design-build-test-learn cycles in bioengineering' has been provisionally accepted for publication in PLOS Computational Biology.

Best regards,

Ruth E Baker

Section Editor

PLOS Computational Biology

Ruth Baker

Section Editor

PLOS Computational Biology

Reviewer's Responses to Questions

**Comments to the Authors:**

Reviewer #2: The authors have satisfactorily addressed all the points raised. The rationale for the methods and approaches

adopted in the study are now much clearer for the reader, and the work as it is presented is a relevant contribution to the community.

**Have the authors made all data and (if applicable) computational code underlying the findings in their manuscript fully available?**

Reviewer #2: Yes

PLOS authors have the option to publish the peer review history of their article (what does this mean?). If published, this will include your full peer review and any attached files.

Reviewer #2: No

---

## [Editor Report · Acceptance letter]

27 Feb 2024

PCOMPBIOL-D-23-01286R1 

teemi: An open-source literate programming approach for iterative design-build-test-learn cycles in bioengineering

Dear Dr K Jensen,

I am pleased to inform you that your manuscript has been formally accepted for publication in PLOS Computational Biology. Your manuscript is now with our production department and you will be notified of the publication date in due course.

With kind regards,

Anita Estes
